# The Resistome of ESKAPEE Pathogens in Untreated and Treated Wastewater: A Polish Case Study

**DOI:** 10.3390/biom12081160

**Published:** 2022-08-21

**Authors:** Jakub Hubeny, Ewa Korzeniewska, Sławomir Ciesielski, Grażyna Płaza, Monika Harnisz

**Affiliations:** 1Department of Water Protection Engineering and Environmental Microbiology, Faculty of Geoengineering, University of Warmia and Mazury in Olsztyn, 10-720 Olsztyn, Poland; 2Department of Environmental Biotechnology, Faculty of Geoengineering, University of Warmia and Mazury in Olsztyn, 10-709 Olsztyn, Poland; 3Environmental Microbiology Unit, Institute for Ecology of Industrial Areas, 40-844 Katowice, Poland

**Keywords:** antibiotic resistance, antibiotic resistance genes, ESKAPEE, wastewater, river water

## Abstract

The aim of this study was to quantify ESKAPEE bacteria, genes encoding resistance to antibiotics targeting this group of pathogens, as well as integrase genes in municipal wastewater and river water. Environmental DNA was extracted from the collected samples and used in deep sequencing with the Illumina TruSeq kit. The abundance of bacterial genera and species belonging to the ESKAPEE group, 400 ARGs associated with this microbial group, and three classes of integrase genes were determined. A taxonomic analysis revealed that *Acinetobacter* was the dominant bacterial genus, whereas *Acinetobacter baumannii* and *Escherichia coli* were the dominant bacterial species. The analyzed samples were characterized by the highest concentrations of the following ARGs: *bla*_GES_, *bla*_OXA-58_, *bla*_TEM_, *qnr*B, and *qnr*S. *Acinetobacter baumannii*, *E. coli*, and genes encoding resistance to β-lactams (*bla*_VEB-1_, *bla*_IMP-1_, *bla*_GES_, *bla*_OXA-58_, *bla*_CTX-M_, and *bla*_TEM_) and fluoroquinolones (*qnr*S) were detected in samples of river water collected downstream from the wastewater discharge point. The correlation analysis revealed a strong relationship between *A. baumannii* (bacterial species regarded as an emerging human pathogen) and genes encoding resistance to all tested groups of antimicrobials. The transmission of the studied bacteria (in particular *A. baumannii*) and ARGs to the aquatic environment poses a public health risk.

## 1. Introduction

Antibiotics play a fundamental role in the treatment of bacterial infections. However, the spread of antibiotic resistance in bacterial communities has led to a steady decline in the efficacy of antimicrobials [1,2]. In recent decades, antibiotic resistance has increased at an alarming rate, and experts have warned that humanity is approaching a post-antibiotic era [3,4]. Until recently, healthcare facilities have been regarded as the main sources of antibiotic resistance, but research has shown that animal husbandry, aquaculture, and urban environments also contribute to the spread of drug resistance. Wastewater treatment plants (WWTPs) pose a particular threat because they discharge antibiotic-resistant bacteria (ARB), antibiotic resistance genes (ARGs), and residual concentrations of antibiotics into downstream aquatic and terrestrial environments [5,6,7].

ESKAPEE (*Enterococcus faecium*, *Staphylococcus aureus*, *Klebsiella pneumoniae*, *Acinetobacter baumannii*, *Pseudomonas aeruginosa*, *Enterobacter* spp., and *Escherichia coli*) bacteria are increasingly associated with antibiotic resistance, which poses a serious threat to public health [8,9]. These bacteria harbor genes that encode resistance to the most popular classes of antibiotics, including vancomycin, methicillin, broad-spectrum β-lactams, carbapenems, fluoroquinolones, and aminoglycosides [10,11,12]. In 2019, the European Center for Disease Prevention and Control (ECDC) published a report on antibiotic resistance in ESKAPEE pathogens in the European Union [13]. According to the literature, an increase in the percentage of vancomycin-resistant enterococci (VRE) harboring the vanA gene is of particular concern [14]. *Staphylococcus aureus*, the second alarming pathogen on the list, can be represented by two genotypes: methicillin-resistant *S. aureus* (MRSA) harboring the mecA gene, and vancomycin-resistant *S. aureus* (VRSA) harboring the vanA gene [15,16]. In recent years, *Klebsiella pneumoniae* has also acquired a variety of β-lactamase enzymes (including bla_NDM_ and bla_KPC_ genes), and this pathogen is now capable of breaking down the chemical structure of penicillins, cephalosporins, as well as carbapenems, which are regarded as drugs of last resort [17,18]. *Acinetobacter baumannii* is an emerging human pathogen that can easily acquire and transfer ARGs to other bacteria [19]. This pathogen harbors Ambler class B and D genes that confer resistance to carbapenems, and it poses a significant problem in the medical sector [20,21,22]. The ESKAPEE group also contains *Pseudomonas aeruginosa*, *Enterobacter* spp., and *E. coli* pathogens that harbor extended-spectrum β-lactamase (ESBL) genes, including subtypes of bla_CTX-M_, bla_SHV_, and bla_TEM_ genes, as well as carbapenem resistance genes, and fluoroquinolone resistance genes from the qnr group [23].

In 2017, the World Health Organization (WHO) published a list of antibiotic-resistant priority pathogens that pose a serious threat to human health, and all ESKAPEE bacteria were placed on that list. Research studies conducted in recent years have demonstrated that ESKAPEE pathogens are transmitted not only in healthcare facilities, but also in environments subjected to anthropogenic and agricultural pressure, including untreated and treated wastewater [24]. In Poland, the spread of ESKAPEE bacteria has been investigated mainly in the hospital sector and in clinical isolates [25,26,27,28,29,30], whereas far fewer studies have examined the prevalence of these pathogens in municipal wastewater and in the natural environment, including river water [31,32].

In view of the above, this metagenomic study was undertaken to quantify ESKAPEE bacteria, genes encoding resistance to antibiotics targeting this microbial group, as well as integron-integrase genes in municipal wastewater and river water. Global surveillance of these pathogens and ARGs has emerged as a critical research problem in an era of growing antimicrobial resistance.

## 2. Materials and Methods

### 2.1. Research Site and Sampling

Environmental samples were collected from a WWTP located in the region of Warmia and Mazury in northeastern Poland (Europe). The analyzed plant is equipped with mechanical and biological wastewater treatment systems. The plant processes wastewater generated in the city of Olsztyn (regional capital) and four surrounding municipalities, and it has an average daily processing capacity of 35,000 m^3^. The plant also receives wastewater from four hospitals, which accounts for 2% of the treated wastewater. Detailed information about the examined WWTP was previously provided by Hubeny et al. [33].

Samples of untreated wastewater (UWW), wastewater exiting from a primary clarifier (PC), wastewater from an activated sludge (AS) bioreactor, wastewater from a multifunctional reactor (MR), treated wastewater (TWW), and river water collected upstream (URW) and downstream (DRW) from the wastewater discharge point were collected in June and November 2018, and in March 2019 (Appendix A). The sampling sites have been described in detail by Rolbiecki et al. [34].

Samples of wastewater and river water were collected in sterile 500 mL and 1000 mL bottles (SIMAX). Wastewater was sampled in triplicate at hourly intervals for 24 h to obtain representative samples, and the collected hourly samples were pooled into a composite sample. River water samples were also collected in triplicate and pooled into a composite sample. The samples were transported to the laboratory at a temperature of 4 °C.

### 2.2. DNA Extraction and Sequencing

The samples were passed through polycarbonate membrane filters (porosity, 0.2 μm) (Millipore, Merck, Darmstadt, Germany) with the use of a vacuum pump (Millipore, Merck, Darmstadt, Germany). The filters were placed in sterile 10 mL Falcon tubes (Eppendorf, Hamburg, Germany), and 5 mL of 1X PBS (Invitrogen, Thermo Fisher Scientific, USA) solution was added. The samples were shaken for 5 h at 50 rpm at room temperature (20–22 °C) using a Grant-Bio PTR-60 rotator (Grant Instruments, Shepreth, UK). The resulting suspension was transferred to sterile 2-mL microcentrifuge tubes (Eppendorf, Hamburg, Germany) and centrifuged (Centrifuge 5415R, Eppendorf, Hamburg, Germany) for 5 min at 4300 *g* at a temperature of 4 °C. The supernatant was discarded, and the obtained pellet was used for DNA isolation. Environmental DNA was extracted from wastewater and river water samples with the Power Water kit (MoBio Laboratories Inc., Carlsbad, CA, USA) according to the manufacturer’s instructions. The volumes of the analyzed samples were: UWW, 40 mL; PC, 40 mL; AS, 20 mL; MR, 40 mL; TWW, 200 mL; URW, 400 mL; and DRW, 400 mL. Samples were quantified via the PicoGreen assay using a VICTOR3 fluorometric plate reader, and their quality and quantity were assessed. Illumina TruSeq DNA PCR-Free libraries were prepared manually based on the manufacturer’s protocol (TruSeq DNA PCR-Free Sample Preparation Guide, Part #15036187 Rev. D; Illumina, San Diego, CA, USA). The libraries were tested using LightCycle qPCR. Library size distribution was assessed using the Agilent Technologies 2100 Bioanalyzer with a DNA 1000 chip, and the libraries were sequenced in the NovaSeq 6000 system in an S4 flow cell lane with 2 × 150 bp configuration.

### 2.3. Bioinformatic Analysis

Wastewater and river water samples collected over three seasons were subjected to metagenomic analysis. FASTQ sequences were quality-filtered, and adaptor sequences were trimmed with Trimmomatic v.0.39 [35] with default parameters. Paired-end sequences were merged using PANDAseq [36]. The reads that were orphans due to quality trimming were excluded from further analyses. The quality of the resulting data sequences was checked using the FASTX-Toolkit 0.0.14–5 (http://hannonlab.cshl.edu/fastx_toolkit/) (accessed on 4 March 2022) with the FASTQ_quality_filter tool. Data were filtered, and reads with less than 80% of bases and a minimum quality score of q = 20 were eliminated. Using seqtk (https://github.com/lh3/seqtk) (accessed on 4 March 2022), FASTQ files were converted to FASTA files for BLAST alignment. Subsequently, sequences were analyzed by BLASTx-type search against the Comprehensive Antibiotic Resistance Database (CARD) [37] (https://card.mcmaster.ca; October 2019 3.0.7) (accessed on 4 March 2022). Samples were parsed and filtered to collect the top hits for each sample, and positive-hit reads with *E* values of ≤1 × 10^−10^ [38], amino acid identities of ≥90% [39], and amino acid alignment lengths of ≥25 were considered [40,41,42,43].

The integron database was created by downloading DNA sequences from INTEGRALL (https://integrall.bio.ua.pt) (accessed on 4 March 2022). Integron sequences were identified by aligning the sequences against database sequences using BLASTn with an *E* value of ≤1 × 10^−10^ [44]. If a BLASTn hit for the alignment against the INTEGRALL database had a ≥90% nucleotide read identity for ≥100 bp, the sequence was annotated as a resistance gene. Sequencing results were recorded in FASTQ files and uploaded to the MetaGenome Rapid Annotation Subsystems Technology (MG-RAST) server for analysis. Each file underwent quality control, which involved quality filtering (removing sequences with five or more ambiguous base pairs) and length filtering (removing sequences with lengths at least two standard deviations from the mean). The identified rRNA sequences were clustered with the UCLUST algorithm [45]. Based on the Greengenes reference database, a representative sequence of each cluster was used for taxonomic identification. Metagenomic data were deposited in the NCBI Sequence Read Archive (SRA) under accession number SRP286056. The results obtained for the samples collected over the three seasons were averaged for further analyses.

The co-occurrence patterns of microbial communities and resistomes were determined by calculating all pairwise Spearman’s rank correlation coefficients (*p*-value < 0.01) [46]. The resulting correlation matrices were translated into an associated network using Gephi 0.9.1 [47].

## 3. Results and Discussion

A resistome analysis was conducted on 15 wastewater samples collected during subsequent stages of treatment, as well as six samples of river water collected upstream and downstream from the wastewater discharge point. Illumina deep sequencing was used to identify ARGs, mobile genetic elements (MGEs), and the taxonomic structure of pathogens in wastewater and river water samples. More than 290 Gb of raw data was generated from 16 samples, and the number of reads ranged from 75.5 million to 201.9 million. Detailed information is presented in Appendix A. The abundance of bacterial genera and species belonging to the ESKAPEE group, 400 ARGs associated with this microbial group, and three classes of integrase genes were determined in metagenomic analyses. The results were averaged and are expressed in terms of gene copies per 1 million reads (ppm).

### 3.1. Taxonomic Structure Analysis

In all analyzed samples, the structure of the microbial communities was determined by assigning reads to taxonomic ranks. Special attention was paid to ESKAPEE pathogens, and the analysis was based on information about bacterial genera and species belonging to this group (Appendix A). The samples collected in the first two stages of wastewater treatment (UWW and PC) were characterized by a predominance of bacteria belonging to the genus *Acinetobacter* (111.5 ppm in UWW, 78 ppm in PC), and the abundance of the remaining bacterial genera (*Enterococcus*, *Pseudomonas*, *Klebsiella*, *Escherichia*, *Enterobacter*, and *Staphylococcus*) was determined in the range of 0–5.4 ppm (Figure 1).

ESKAPEE pathogens belong to the phyla Firmicutes and Proteobacteria, which are the most prevalent bacterial groups in wastewater. The predominant bacterial families are Enterococcaceae (including *Enterococcus* spp.) of the phylum Firmicutes, as well as Enterobacteriaceae (including *Klebsiella* spp. and *Escherichia* spp.) and Moraxellaceae (including *Acinetobacter* spp.) of the phylum Proteobacteria [48].

An analysis of six ESKAPEE group species (*Acinetobacter baumannii*, *Enterococcus faecium*, *Escherichia coli*, *Klebsiella pneumoniae*, *Pseudomonas aeruginosa*, and *Staphylococcus aureus*) confirmed that *A. baumannii* (1.16 ppm in UWW; 1.62 ppm in PC) and *E. coli* (1.16 ppm in UWW; 1.62 ppm in PC) were the dominant pathogens. In successive stages of wastewater treatment (samples AS and MR), a significant reduction was observed in the abundance of ESKAPEE genera (0.002–0.64 ppm and 0–0.3 ppm, respectively) and species (0–0.014 ppm and 0–0.015 ppm, respectively).

Similar results have been reported by other authors. Numberger et al. [48] analyzed the abundance of the three most prevalent bacterial phyla (Proteobacteria, Bacteroidetes, and Firmicutes) in samples of UWW and TWW, and found that bacteria of the family Moraxellaceae were more prevalent than bacteria of the family Enterobacteriaceae. Furthermore, *Acinetobacter* was the dominant bacterial genus in a taxonomic structure analysis conducted by McLellan et al. [49].

The activated sludge process in the WWTP equipped with a biological and mechanical wastewater treatment system reduced the abundance of ESKAPEE genera and species by 80% on average (UWW vs. TWW) (Appendix A). In other studies, biological and mechanical wastewater treatment systems led to a similar reduction in microbial levels (80–99%) [33,50]. According to the literature, the effectiveness of wastewater treatment can be significantly affected by the type of wastewater inflow and fluctuations in a plant’s performance [51,52,53]. There is evidence to indicate that the activated sludge process can both decrease and increase the size of the bacterial population in wastewater. The results obtained for bacterial genera and species indicate that their abundance decreased drastically during successive stages of biological treatment. This is not surprising because the biological treatment process is expected to reduce microbial abundance by up to 99.99%. However, an analysis of treated effluent samples revealed an increase in microbial abundance. The mechanisms underlying this phenomenon have not been thoroughly described in the literature. Nonetheless, it appears that it could be caused by the immobilization of heavy metals or antibiotics (factors exerting selective pressure) in sewage sludge. Another reason could be the formation of biofilm in successive stages of biological treatment. All processes responsible for changes in bacterial abundance during wastewater treatment have not yet been fully elucidated, but an analysis of the pattern of these changes may shed light on the activity of microbial communities.

An analysis of bacterial genera and species in river water sampled downstream from the wastewater discharge point revealed an increase in the counts of two genera and their associated species. The abundance of *A. baumannii* increased from 0.000 ppm in URW to 0.007 ppm in DRW, whereas *E. coli* counts increased from 0.008 ppm to 0.035 ppm (Figure 2). The release of treated effluent into the river had no effect on the abundance of the remaining bacterial species. According to the literature, WWTPs are potential sources of both *E. coli* [54,55] and *A. baumannii* [56] in river water.

A comparison of microbial abundance at the level of genus and species in samples collected in different seasons and at different sites did not reveal any clear trends. However, a comparison of the results obtained during the entire treatment process shows that the average abundance of the studied taxonomic groups was highest in November, followed by June and March (Appendix A). The above trend was also observed in an earlier study by Hubeny et al. [57], where the number of *Acinetobacter* sp. strains isolated from wastewater and river water was highest in fall and lowest in winter.

### 3.2. Prevalence of Antibiotic Resistance Determinants

The prevalence of ARGs and class 1, 2, and 3 integron-integrase genes was determined with the use of CARD and INTEGRALL databases. Four hundred ARGs (including subtypes), including *bla*_PER_, *bla*_VEB_, *bla*_IMP_, *bla*_VIM_, *bla*_GES_, *bla*_KPC_, *bla*_NDM_, *bla*_OXA_, *bla*_CTX-M_, *bla*_SHV_, and *bla*_TEM_ (encoding resistance to β-lactams); *qnr*B, *qnr*C, *qnr*D, and *qnr*S (encoding resistance to fluoroquinolones); *aph(2’’)-If, aph(2’’)-Ig, aph(2’’)-Ia,* and *aac(3)-IV* (encoding resistance to aminoglycosides); as well as *int*I1, *int*I2, and *int*I3 integrase genes were selected for analysis. The selected genes confer resistance to the antibiotic groups listed in the ECDC report on antimicrobial resistance in ESKAPEE pathogens. Detailed information about the analyzed gene types and subtypes is presented in the Appendix A.

The examined ARGs were most prevalent in UWW (0.009–2.36 ppm) and PC (0.005–2.34 ppm) (Figure 3). In the work of Karkman et al. [58], ARGs were also most prevalent in samples collected in the initial stages of wastewater treatment. In the present study, the following genes were predominant in UWW and PC: *bla*_GES_ (2.36 and 2.34 ppm, respectively), *bla*_OXA-58_ (1.82 and 1.93 ppm, respectively), *bla*_TEM_ (0.72 and 0.76 ppm, respectively), *qnr*B (0.25 and 0.36 ppm, respectively), and *qnr*S (1.08 and 1.45 ppm, respectively) (Appendix A). These genes were also frequently identified in wastewater in other studies. Wang et al. [59] found that *bla*_TEM_, *bla*_OXA_, and *qnr*S were highly prevalent in municipal water in Europe, USA, and Canada. Pärnänen et al. [60] investigated the prevalence of ARGs in Europe and reported that *bla*_GES_ and *bla*_OXA_ were commonly detected in influent and effluent wastewater. In the current study, genes encoding resistance to β-lactams and quinolones were most prevalent in UWW and PC. According to the ECDC report for 2020, β-lactams and quinolones were widely used in the Polish medical sector [61]. Research has demonstrated that the use of specific antibiotics in healthcare facilities is frequently correlated with an increase in the prevalence of ARGs in wastewater [62]. A decrease in the abundance of the tested genes was observed when TWW was compared with DRW. Tang et al. [63] also noted a decrease in the abundance of ARGs when comparing TWW with river water sampled downstream from the wastewater discharge point. This phenomenon can be explained by the fact that wastewater reaching the river can be diluted more than 250 times. Natural self-purification of water bodies also contributes to decreasing the abundance of ARGs in river water samples. The absence of the *bla*_PER-2_ gene in DRW can be explained in a similar way.

The heat map revealed a significant decrease in ARG abundance in TWW. However, the prevalence of ARGs in TWW increased relative to the samples collected from the activated sludge reactor (AS) and the multifunctional reactor (MR) (Figure 3); however, the observed differences were not statistically significant (Appendix A). In a study by Zieliński et al. [64], the abundance of ARGs encoding resistance to β-lactams and quinolones was also higher in TWW than in the AS. During subsequent stages of biological treatment, the abundance of ARGs decreases due to reduced nutrient availability [65]. In turn, an increase in the prevalence of antibiotic resistance determinants in TWW may be associated with the formation of biofilm by bacteria harboring those genes [64]. Despite the observed increase in ARG abundance, more than 90% of the examined genes were removed during the biological treatment process (Appendix A). In other studies, the prevalence of ARGs was also reduced in TWW [66,67,68].

An analysis of integrase genes (MGEs) revealed that *int*I1 was more prevalent in wastewater samples (78.17–2911.29 ppm) than *int*I2 (0.37–34.41 ppm) and *int*I3 (0.15–10.03 ppm) (Figure 4 and Appendix A). Similar results were noted in our previous study [33], where *int*I1 was the most abundant integrase gene (166.59–2883.91 ppm), whereas *int*I2 (0.02–29.87 ppm) and *int*I3 (0.09–2.46 ppm) were determined in smaller quantities. The concentrations of integrase genes followed a similar pattern in other studies [69,70].

The present study confirms that the analyzed WWTP contributed to the contamination of river water with ARGs and all analyzed integrons. The developed heat maps indicate that the abundance of *bla*_VEB-1_, *bla*_IMP-1_, *bla*_GES_, *bla*_OXA-58_, *bla*_TEM_, *qnrS*, and class 1, 2, and 3 integron-integrase genes was higher in river water sampled downstream than upstream from the wastewater discharge point (Appendix A). Sabri et al. [71] reported similar results in samples of river water collected downstream from the wastewater discharge point. In the cited study, ARG concentrations were also higher in sampling sites located downstream than upstream. The concentrations of ARGs and class 1 integron-integrase genes were more than two orders of magnitude higher in samples collected from downstream sites. Czekalski et al. [72] and Osińska et al. [73] also reported an increase in the abundance of antibiotic resistance determinants in river water receiving effluents from WWTPs. In the present study, effluents from four hospitals accounted for 2% of the wastewater processed by the analyzed WWTP. According to Hocqeut et al. [74] and Paulus et al. [75], hospitals play a major role in the transmission of ARB and ARGs. The polluting effects of hospital wastewater could be exacerbated by the fact that the use of last-resort drugs has increased in recent years, and hospital effluents could have a different ARG profile than other types of wastewater. This observation indicates that growing levels of antibiotic resistance and the spread of ARGs in the environment can be largely attributed to human activity [76].

Aquatic environments such as wastewater and river water are affected by physicochemical factors that contribute to changes in these environments [77]. Zieliński et al. [78] focused on the epidemiological threat posed by antibiotic resistance determinants to WWTP employees and the environment. The cited authors studied the same samples of wastewater, sewage sludge, and river water as those analyzed in the current study, and described the influence of physicochemical parameters on the prevalence and diversity of ARGs. Based on their data, it can be concluded that total organic carbon (TOC), chemical oxygen demand (COD), and biological oxygen demand (BOD) are strongly correlated with the concentrations of different groups of ARGs, in particular, those conferring resistance to β-lactams. According to the literature, lower TOC levels contribute to an increase in the diversity and abundance of ARGs. Environmental physicochemical parameters could have a direct effect on the diversity and the abundance of ARGs in water bodies.

A comparative analysis of ESKAPEE abundance in different seasons did not reveal any specific trends of changes in their concentrations in samples collected during subsequent stages of the treatment process. A comparison of overall ARG abundance data between seasons revealed only negligible differences (Appendix A).

The extent to which ARB and ARGs present in the natural environment affect human health continues to attract increasing interest from researchers. Bouki et al. [79] reported an increase in the number of environmental strains harboring ARGs that are most often detected in clinically relevant bacteria, which implies that many bacteria have developed resistance to one or more antimicrobials. Ultimately, these bacteria can be transferred from the environment to humans.

### 3.3. Correlation Analysis

The examined bacterial species of the ESKAPEE group, ARGs, and integrase genes were subjected to a correlation analysis. The number and strength of the correlations between the studied variables were determined in a network analysis (Figure 5). Three types of correlations were investigated: (1) bacterial species vs. ARGs, (2) bacterial species vs. integrase genes, and (3) integrase genes vs. ARGs. In all analyses, the values of the correlation coefficient ranged from 0.875 to 0.964 (Appendix A). The results of Spearman’s rank-order correlation tests are presented in the Appendix A. Not all of the examined ARGs were bound by significant correlations with ESKAPEE species or integrase genes. However, a correlation analysis investigates the relationships between quantitative variables; therefore, the absence of significant relationships could be due to the fact that some genes occur rarely in a given environment [80].

In the group of the analyzed bacterial species, *A. baumannii* and *E. faecium* were bound by the highest number of significant correlations with ARGs. The presence of these species was associated with the prevalence of genes encoding resistance to β-lactams, aminoglycosides, and fluoroquinolones. A network analysis conducted by Alexander et al. [81] revealed significant correlations between *A. baumannii* and enterococcus counts and the abundance of genes encoding resistance to β-lactams in wastewater samples. Strong correlations between microbial counts and the above ARGs could suggest that: (1) the presence of these bacteria in the group of opportunistic ESKAPEE pathogens is associated with hospital infection outbreaks, and (2) extensive use of β-lactams and fluoroquinolones in the medical sector contributes to antibiotic resistance in the environment [82,83,84].

Integrons are MGEs that can harbor ARGs [85]. Integrons and other MGEs (such as transposons and plasmids) have a high potential for transferring ARGs [86]. The role of integrons as potential hosts for ARGs has been analyzed in the literature [87,88]. In the current study, integrase genes were also bound by significant correlations with ARGs. The *int*I3 gene was strongly correlated with *bla*_GES_ and *bla*_VEB-1_, as well as with *int*I1 and *int*I2 genes. Both *int*I1 and *int*I2 were associated with *APH(2’’)-Ia*, *bla*_CTX-M_, *bla*_PER-1_, and *bla*_VEB-1_ genes, as well as *A. baumannii* counts. *Acinetobacter baumannii* was the only ESKAPEE species that was correlated with integrase genes. According to the literature, the genome of *A. baumannii* harbors both class 1 [89,90,91] and class 2 integrons [92,93,94]. Class I integrons can carry 40 ARGs associated with aminoglycosides, β-lactams, chloramphenicol, macrolides, and sulfonamides [91]. The absence of a direct correlation between *A. baumannii* and *int*I3 could be attributed to the fact that this integron is very rarely found in the genome of the above bacterial species [95,96]. In recent years, *Acinetobacter* spp. (including *A. baumannii*) have been identified as a major cause of nosocomial infections, and these emerging human pathogens have impressive genetic capabilities for acquiring and disseminating antibiotic resistance. *Acinetobacter baumannii* strains isolated from hospitals and the natural environment were positively correlated with ARG abundance [97]. The results of the present study and the findings of other authors suggest that the spread of ESKAPEE pathogens, in particular *A. baumannii*, in river water can increase the prevalence of antibiotic resistance determinants (integrons) and expand the resistome of environmental bacteria.

## 4. Conclusions

This study investigated the prevalence of opportunistic pathogens from the ESKAPEE group and antibiotic resistance determinants in wastewater and river water samples. The examined pathogens and genes determining resistance to clinically important antimicrobials were identified in the studied samples. These results indicate that WWTPs play an important role in the dissemination of critical ARGs and ESKAPEE bacteria in the natural environment. Despite the fact that the abundance of the analyzed bacteria was significantly reduced by the wastewater treatment process, river water sampled downstream from the wastewater discharge point was polluted with *A. baumannii* and *E. coli*, which are important pathogens with highly mobile resistomes. A correlation analysis revealed that integrase genes were strongly correlated only with *A. baumannii*, which has been identified as an emerging human pathogen. The presence of these bacteria in the aquatic environment poses a serious threat to public health. To the best of our knowledge, this is the first study to investigate the transmission of ESKAPEE pathogens and ARGs from the wastewater matrix to the natural environment in central–eastern Europe. Further research is needed to elucidate the evolution of antibiotic resistance mechanisms in clinically important ESKAPEE bacteria in the natural environment.

## Figures and Tables

**Figure 1 biomolecules-12-01160-f001:**
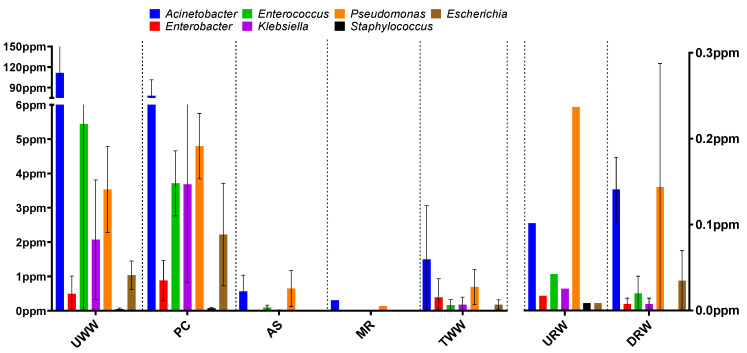
Abundance of ESKAPEE group members in the analyzed samples (genus level). The distribution of data (standard deviation) is marked by whiskers in the box plot. Sampling sites: UWW—untreated wastewater, PC—wastewater exiting from the primary clarifier, AS—wastewater from the activated sludge tank, MR—wastewater from the multifunctional reactor, TWW—treated wastewater, URW—river water collected upstream from the wastewater discharge point, DRW—river water collected downstream from the wastewater discharge point. Data regarding UWW, PC, AS, MR, and TWW correspond to the left axis, and data regarding URW and DRW correspond to the right axis.

**Figure 2 biomolecules-12-01160-f002:**
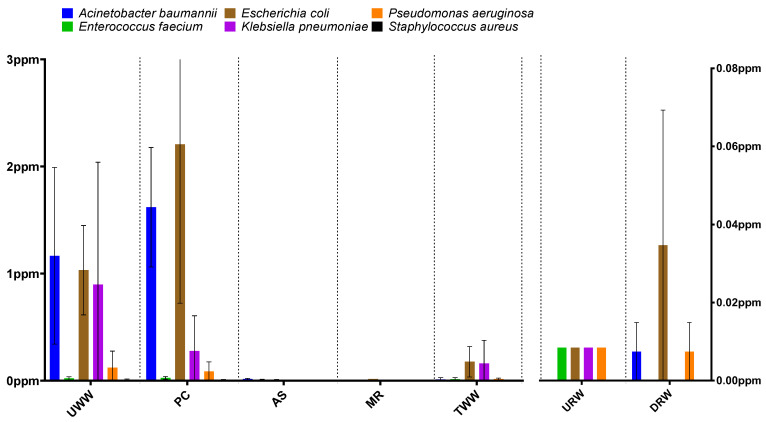
Abundance of ESKAPEE group members in the analyzed samples (species level). The distribution of data (standard deviation) is marked by whiskers in the box plot. Sampling sites: UWW—untreated wastewater, PC—wastewater exiting from the primary clarifier, AS—wastewater from the activated sludge tank, MR—wastewater from the multifunctional reactor, TWW—treated wastewater, URW—river water collected upstream from the wastewater discharge point, DRW—river water collected downstream from the wastewater discharge point. Data regarding UWW, PC, AS, MR, and TWW correspond to the left axis, and data regarding URW and DRW correspond to the right axis.

**Figure 3 biomolecules-12-01160-f003:**
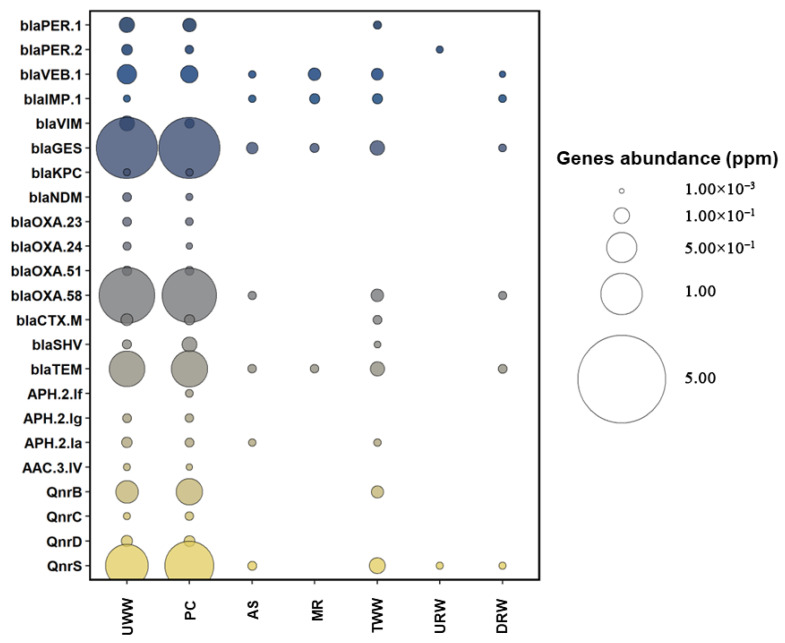
Distribution of antibiotic resistance genes (ppm) in subsequent stages of wastewater treatment and in river water. Sampling sites: UWW—untreated wastewater, PC—wastewater exiting from the primary clarifier, AS—wastewater from the activated sludge tank, MR—wastewater from the multifunctional reactor, TWW—treated wastewater, URW—river water collected upstream from the wastewater discharge point, DRW—river water collected downstream from the wastewater discharge point.

**Figure 4 biomolecules-12-01160-f004:**
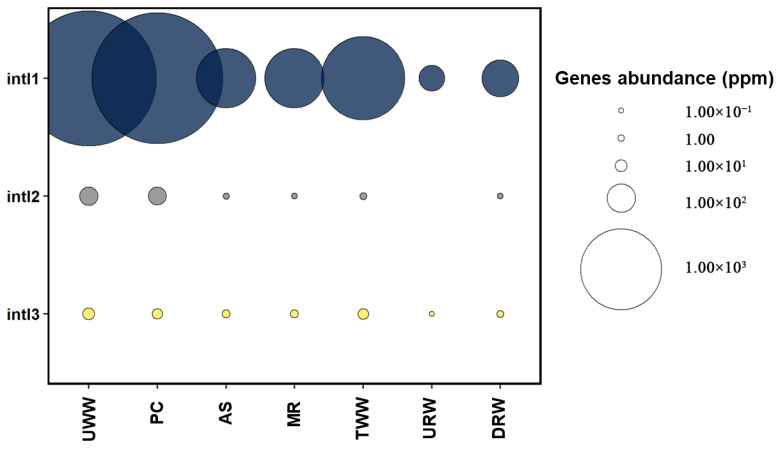
Distribution of integron-integrase genes (ppm) in subsequent stages of wastewater treatment and in river water. Sampling sites: UWW—untreated wastewater, PC—wastewater exiting from the primary clarifier, AS—wastewater from the activated sludge reactor, MR—wastewater from the multifunctional reactor, TWW—treated wastewater, URW—river water collected upstream from the wastewater discharge point, DRW—river water collected downstream from the wastewater discharge point.

**Figure 5 biomolecules-12-01160-f005:**
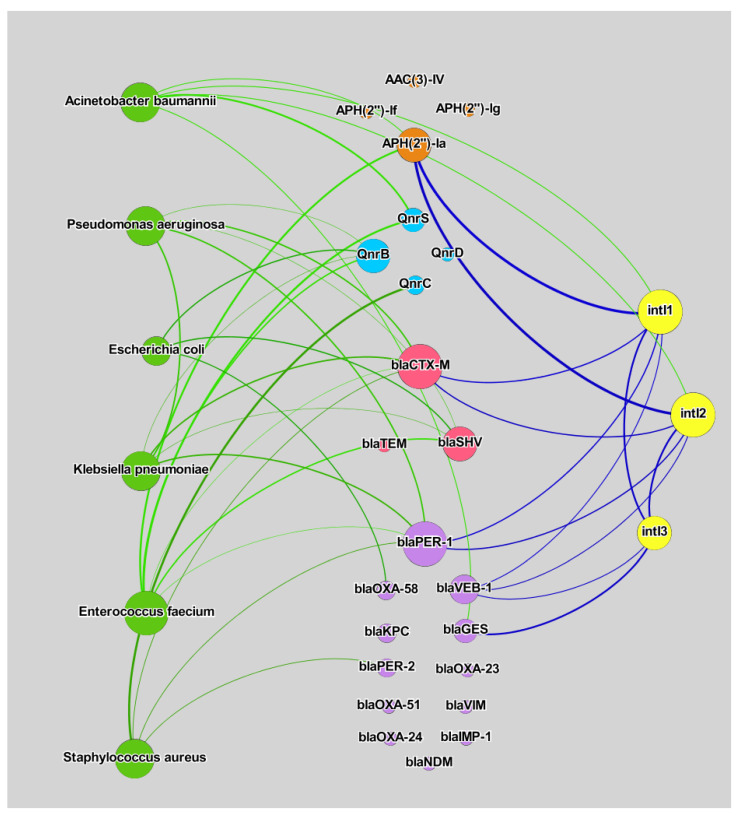
Network analysis of the correlations between variables in wastewater and river water samples. Three types of correlations were evaluated: (1) bacterial species vs. ARGs, (2) integrase genes vs. ARGs, and (3) bacterial species vs. integrase genes. All correlations (edges) in the analysis are statistically significant.

## Data Availability

Metagenomic data were deposited in the NCBI Sequence Read Archive (SRA) under accession number SRP286056.

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
