# Peer review of "The Resistome of ESKAPEE Pathogens in Untreated and Treated Wastewater: A Polish Case Study"

_biomolecules, 2022, doi:10.3390/biom12081160_

Round 1

Reviewer 1 Report

This paper represents ESKAPEE (in general ESKAPE) bacteria and antibiotic related genes such as bla and MGEs in water samples collected from each process of a wastewater treatment plant and upstream and downstream of the river, based on sequencing analysis of DNA. The experimental method is orthodox. The analysis method, the arrangement, and opening of the obtained data are correctly executed. Therefore, it is judged that the reliability of this paper is guaranteed. New information, findings on the so-called ESKAPE, and related genes that have been obtained.

However, important information and analysis are missing in the following two items.

(1) Since this is a study comparing changes in bacteria and genes in the wastewater treatment process, it is essential to show the basic water quality items in each treatment process. The authors have already published a number of related achievements, and it is speculated that pH, EC, BOD (COD or TOC), fecal indicator bacterial counts (E. coli, coliform bacteria, enterococci) have been investigated.

(2) All the analysis results of samples with different sampling dates are summarized and comprehensively discussed. Influent wastewater and treatment processes at wastewater treatment plants vary depending on the month and seasonal differences of sampling. Since you have conducted three different surveys, it is necessary to compare at least the changes in the bacterial species and resistance genes of each sampling.

Since this paper provides information on the basis for the release of pathogenic bacteria and drug resistance-related genes from sewage treatment plants into the river environment, I support the publication after appropriate revision.

Some important points that need to be considered are listed as follows:

1) ESKAPEE

Please explain why you chose E. faecium and why you removed the main species, E. faecalis. In addition, there are other important types of Enterobacter. Based on the results of deep sequencing analysis, a comprehensive analysis is possible for the other major bacteria and flora.

2) Line 104–105

Isn't DNA recovered by a membrane filter? A brief explanation should be needed.

3) It is necessary to explain why “Abundance” is higher in the final treated water than that in the treated water of the biological process. This question is common to the results of Figures 1, 2, and 3.

4) Fig. 3

The consideration the reason why the gene blaPER.2 detected at the upstream river is not detected at the downstream is needed. In addition, the reason why the genes contained in the final treated water were not detected at the downstream point where the treated water was discharged is also important.

5) Line 289–293

The authors insist on the possibility of a route from the environment to humans. Even if the same resistant strain or resistance gene is detected in the environment and the hospital (human), the starting point remains unknown. I also agree with your consideration and risk proposal.

6) Line 297­–29, Fig. 5

Why is the important Enterobacter in coliform bacteria not shown in Fig.5?

Reviewer 2 Report

The authors present a fairly comprehensive resistome and correlational analysis of ESKAPEE bacteria in a WWTP and receiving river waters in Poland.  The paper reads well and flows logically from design to conclusion with only minor issues to be addressed.  For this reason, this reviewer suggests acceptance with minor revision.  Issues to be addressed are identified below:

1) The paper reads well, but could benefit from one final proof reading for English usage and grammar. Please revise for clarity.

2) Line 118: Reference should be in same format as others.

3) Section 2.3 Bioinformatic Analysis: Section could benefit from a bit of additional description as to why exclusions were made (e.g. lines 118-119).  This will augment reproducibility and continuity across studies. 

4) Line 138: Please provide citation or description of the UNCLUST algorithm for reproducibility and clarity.

5) Lines 148-150: Statement appears to have been a carryover from a manuscript template.  Please delete. 

6) Figures 1 and 2: It is unclear which data correspond with the left vs. right axis scale.  Please clarify.

7) Lines 210-211: This statement could benefit from some further discussion, otherwise it is unnecessary and should be omitted.

8) line 214: You report A. Baumannii increased from 0.00 ppm to 0.007 ppm.  Is the 0.00 ppm really a non-detect? Please report a value other than 0 for the ability to make comparisons between the two sampling sites.

Round 2

Reviewer 1 Report

The revised manuscript has been revised to properly reflect my suggestions and comments.

I recommend the publication of this paper.